# Fading Roars? A Survey of the Cultural Use and Illegal Trade in Wild Felid Body Parts in Côte d’Ivoire

**DOI:** 10.3390/ani15030451

**Published:** 2025-02-06

**Authors:** Robin Horion, Janvier Aglissi, Rob Pickles, Amara Ouattara, Marine Drouilly

**Affiliations:** 1Panthera Senegal, Tambacounda 26500, Senegal; aglissi@hotmail.fr (J.A.); rpickles@panthera.org (R.P.); mdrouilly@panthera.org (M.D.); 2Office Ivoirien des Parcs et Réserves (OIPR), Abidjan 06 BP 426, Côte d’Ivoire; amara.ouattara@oipr.ci; 3Ecole Doctorale Biologie-Environnement, Santé Université Félix Houphouët-Boigny, Abidjan 01 BP 34, Côte d’Ivoire; 4Institute for Communities and Wildlife in Africa (iCWild), University of Cape Town, Cape Town 7700, South Africa; 5Centre for Social Science Research (CSSR), University of Cape Town, Cape Town 7700, South Africa

**Keywords:** Côte d’Ivoire, cultural practices, dozo, felids, féticheur, illegal wildlife trade, marabout, market, traditional medicine, zootherapy

## Abstract

The illegal trade in wild felids is a serious threat to their survival in West Africa, including in Côte d’Ivoire. From April to June 2024, we conducted an exploratory study in 16 towns and villages across Côte d’Ivoire to understand the characteristics and some of the general drivers of the illegal trade in wild cat products. We surveyed 46 markets and interviewed 39 sellers and 14 users of felid products. We discovered that more than 80% of the cities visited had wild felid products for sale, nearly half of the markets were selling leopard parts and 25% were selling lion parts. This trade was mostly driven by the demand for body parts used in traditional medicine and cultural practices, with leopards and lions having high cultural significance. Sellers and users indicated that most products were sourced from northern neighboring countries, even if some countries in other African regions were also mentioned. We recommend continuing to monitor the illegal wildlife trade through the development of regional databases and enhanced market and consumer survey protocols. Additionally, targeted behavioral change campaigns should be implemented to address corruption, increase enforcement agencies’ capacity and interest in tackling the illegal wildlife trade, and shift the practices of traditional practitioners toward sustainable alternatives.

## 1. Introduction

The illegal wildlife trade (IWT) threatens biodiversity globally, and generates billions of dollars annually, ranking it among the most lucrative illegal trades after drugs and arms trafficking [1]. The illegal trade in wildlife not only endangers species but also disrupts entire ecosystems, as the removal of keystone species can lead to cascading effects that destabilize natural habitats and result in ecological collapse [2,3]. Biodiversity loss due to wildlife trafficking has far-reaching consequences, affecting not only animal populations but also human communities that rely on natural resources for livelihoods, food security, and traditional practices, while also increasing the risk of zoonotic disease transmission through close contact with wildlife [4,5]. In West Africa, the majority of research on illegal wildlife trade has focused on the commercial trafficking of products like elephant ivory, pangolin scales, and bushmeat, where financial or nutritional motivations are predominant [6,7,8,9]. However, far fewer studies have investigated the traditional uses of wildlife for cultural, spiritual, social and medicinal purposes, which are locally significant [9]. These traditional practices often involve a different range of species and products – including wild felids – follow distinct trade routes, and respond to a different set of demands [10,11] Wild cats are also sold internationally for financial gain, either as pets or for their parts, with skins frequently seized at borders, en route to the Middle East, Asia, or Europe [12].

In West Africa, the relationship between felids and humans has a rich historical background and strong cultural significance. Leopards (*Panthera pardus*) are prominently depicted in royal iconography, especially among the Akan societies, as seen in bronze sculptures and plaques from the 13th to 19th centuries [13]. West African ancient tales, such as those recorded by Amadou Hampâté Bâ, a Malian writer, historian, and ethnologist, depict lions (*Panthera leo*) and other felids as central, symbolic characters, highlighting their importance in oral traditions [14]. Today, in West Africa and across the continent, felids remain significant; kings and notable figures often wear big cat skins (i.e., lion and leopard) as symbols of reputation, social status, and spiritual protection [15]. Body parts play a key role in cultural ceremonies due to their perceived medicinal and spiritual properties. In West Africa, wild cat parts are widely sold in local markets and are frequently used by traditional healers and spiritual leaders [10,11]. These practices are now deeply embedded in West African cultures, reflecting a syncretism that Christianity, Islam, animism and fetishm [16]. 

Although these practices play a role in preserving traditions and cultural heritage, the pressure they exert on wild cat populations has contributed to their decline both regionally and globally [10,17,18]. Lion and leopard numbers are declining drastically across Africa, due to severe threats [19,20]. In West Africa, the situation is especially dire, with lions listed as Critically Endangered [21] and only present within two populations: Niokolo-Koba National Park (NKNP) in Senegal and the W-Arly-Pendjari (WAP) complex straddling Benin, Burkina Faso and Niger. Leopards are also at risk, with the first IUCN Red List assessment for the species at the regional level highlighting increasing threats and a relatively small and declining population [22]. Smaller cats, although generally more resilient, face an important lack of research in the region and very little is known about their population status [23]. For example, the African golden cat (*Caracal aurata*), classified as vulnerable across its range, has barely been studied and no information on the West African populations (i.e., size, trend, density) is available [24,25]. 

In Côte d’Ivoire, lions likely disappeared around the year 2000 during the political instability affecting Comoé National Park (CNP) [26], whereas leopards still persist in a few protected areas, such as Taï National Park (TNP), Marahoué, and Comoé National Parks [19,27,28]. The African golden cat is reported in TNP and CNP, and there are sparse records for the other small felids: serval (*Leptailurus serval*), caracal (*Caracal caracal*) and African wildcat (*Felis lybica*) [27,28]. In the country, the illegal trade in felids and their parts remains largely undocumented, but it is suspected to be significant due to cultural demand and economic incentives, as suggested by reports such as those from the EAGLE Network [12,29]. This trade could drive the West African populations of these species to the brink, exacerbating local declines and increasing demand from other parts of Africa, putting additional strain on already vulnerable species [30].

Relying solely on law enforcement to counter the illegal wildlife trade is often insufficient, as this approach can drive the trade underground or be hampered by limited resources and corruption [31]. To address these challenges, behavioral change campaigns are emerging as an effective strategy by targeting the socioeconomic factors that influence demand [32]. By understanding the motivations behind user choices and applying a socioeconomic decision-making model [33], these campaigns can effectively shift behavior and reduce reliance on wildlife products. Research indicates that the most successful efforts involve collaboration with local leaders, acknowledging traditional practices, and providing culturally appropriate alternatives [34]. Studies highlight that conservation results improve when local knowledge is integrated and respected [35], especially in areas where wildlife holds spiritual and medicinal significance. Effective campaigns therefore need to balance biodiversity protection with cultural sensitivity, creating solutions that are both impactful and respectful [36].

Our study aimed to explore some of the characteristics of the illegal trade of wild cats in Côte d’Ivoire. To do so, we visited traditional markets between April and May 2024 to identify stalls selling wildlife products, specifically wild cat body parts. We also conducted semi-structured interviews with vendors, users, and traditional practitioners to gain insight into the reported sources of the products, their uses, and the frequency of presence in local markets.

Our study was guided by three primary research questions:How do the relative abundance, diversity, and distribution of wild cat products vary across local markets in Côte d’Ivoire?Can we establish a typology of traditional practitioners using wild felid products and their role in this illegal trade?What are the main reported centers, trade routes, and sources of wild cat products that are illegally sold in local markets?

At the end of this article, we propose practical, culturally informed and timely strategies to counter this illegal trade in wild cat products in the country and beyond.

## 2. Materials and Methods

### 2.1. Study Area and Market Identification Methods

To investigate the illegal trade in wild cat products across Côte d’Ivoire’s main cities, we targeted key urban centers within each region, including those on major trade routes and near border areas. Smaller cities surrounding Comoé National Park were also included in the study because we hypothesize that there might be wildlife products directly sourced from the park sold in the surrounding towns. Initial identification of major markets where wildlife products are sold was initially informed by a review of the scientific and gray literature, including news and social media, where we looked for the terms “wildlife market” OR “traditional market” OR “medicinal markets” AND “Côte d’Ivoire”, both in French and in English. We also contacted three local and two international researchers by email, who have been involved in illegal wildlife trade research in the country. This initial search was followed by in-country discussions with park authorities, local law enforcement (Eaux et Forêts Ministry) officials, and staff working for NGOs, such as the EAGLE Network, as well as with locals, including hotel staff and taxi drivers. 

### 2.2. Local Guides and Interview Facilitation

Once the main wildlife markets were identified, we recruited local guides in each locality, to assist with contacting sellers and traditional practitioners. Local guides are well-respected individuals in their community or suburb, with extensive knowledge of the city or region where we conducted our research, some of whom are involved in tracking or monitoring illegal wildlife trade. They were contacted and recruited before our visits to the markets, were fully informed about the study, and agreed to assist us by facilitating discussions with sellers or connecting us with traditional practitioners. They explained the purpose of the study in local languages to the people we wanted to interview, helped build rapport and trust, and provided translation when French was not well understood. Local guides played a crucial role in facilitating access to respondents involved in illegal activities, including wildlife trade and use. Direct requests from researchers can be ineffective if sellers and users perceive them as outsiders, or if trust has not been built beforehand [37]. By leveraging local guides—whether they are community leaders or users themselves—we were able to gain the respondents’ trust, discuss sensitive information and expand our network of interviewees. The latter was also achieved through the "snowball sampling" technique, where the initial contacts made in a market introduced us to additional participants, allowing us to locate respondents more efficiently through existing connections and to broaden the scope of our interviews [38].

### 2.3. Observations and Interviews in Wildlife Markets

From April to June 2024, we conducted systematic field observations in the identified markets to document the diversity and abundance of wildlife products, with a particular focus on stalls selling wild cat species [39]. These observations included recording the species on display, the types of products sold, their quantities, and their prices. We also took note of vendor characteristics such as their biological sex, approximate age, ethnic group, and country of origin to better contextualize the trade dynamics. Whenever feasible, informal interviews were conducted with vendors to gather more detailed information about their practices. These discussions provided insights into the potential uses of wild cat products, as well as the broader cultural and economic contexts driving their sale. If vendors consented, we conducted more in-depth interviews after explaining the purpose of the study, which allowed us to delve into the uses of specific body parts and their perceived value (Appendix A). Our positionality as researchers was that of "passive observers", maintaining transparency about the study while avoiding direct engagement in trade practices [33]. Most of the data collected were based on observations, and interviews were relatively rare. Sellers are generally aware that their activities are prohibited, which made it difficult to conducting formal interviews difficult. As a result, many questions, particularly those listed in Appendix A, could not be asked. When interviews did take place, the most frequently asked questions were questions 13 to 20 and 25, depending on the level of cooperation from the sellers.

### 2.4. Interviews of Practitioners

In addition to the market vendors, we interviewed users of wildlife products found through the local guides, including traditional healers (i.e., marabouts: refers to a man with magical or healing powers, or to a Muslim religious leader (like a sheikh), or a Quran teacher for children [40]) in their home or their practice space, and traditional hunters (i.e., Dozo hunters are members of a traditional hunter confraternity from the Manding cultural sphere, known for their deep knowledge of the spiritual and natural worlds [41]). These interviews were conducted to obtain a deeper understanding of the cultural and ritual uses of wild cat body parts. Unlike market interviews, these discussions were pre-arranged and loosely followed an interview guide (Appendix A). They provided valuable insight into the demand for illegal wild cat products. However, not all questions were systematically asked, as the flow of the interview depended on the available time and the level of cooperation from the interviewees. As a result, some sensitive questions, particularly those concerning forbidden practices (such as illegal trade), could either not be asked or the responses was discarded when the interviewee was uncooperative or too evasive (see Section 2.6 data reliability and analyses).

### 2.5. Ethical Considerations and Participant Consent

Before conducting any interviews, we obtained verbal consent from the participants, after explaining the research project and answering their questions. For participants who were illiterate, the field researcher signed the consent form on their behalf with an "X" and a serial number. When permitted, interviews were audio recorded and securely stored in a password-protected folder. The audio files were then transcribed into Word documents, which were similarly stored on a single personal computer. Each interview was assigned a coded identifier to anonymize the participants, ensuring that their names or identifying details could not be traced. In cases where participants declined to be recorded, manual notes were taken and digitized. All collected data were anonymized, and no identifying information about the participants was collected. Names and addresses were not recorded, although phone numbers were stored separately with the participants’ consent, for any follow-up questions. To acknowledge their contribution and time off work, participants were compensated with modest refreshments or meals after or during the interviews took place. The study was approved by the Ethics Committee of the University of Cape Town (reference CSSR/REC/2024/01_1; date of approval: 12 April 2024) and conducted in accordance with the University of Cape Town Code for Research involving Human Subjects (refer to the “Institutional Review Board Statement” section).

### 2.6. Data Reliability and Analyses

The reliability of information obtained from interviews varies and can lead to misinformation [42]. To mitigate these issues, only first-hand information corroborated by at least one other independent source or enforcement record was included in the analysis [43]. Additionally, information about product sources was corroborated with data on felids from the Illegal Wildlife Trade Portal “https://www.wildlifetradeportal.org (accessed on 11 August 2024)”. After initial coding, all interview transcripts were uploaded to the online version of Taguette (accessible at “https://www.taguette.org (accessed on 7 September 2024)”), a free and open-source text tagging tool for qualitative data analysis and qualitative research. This software was chosen due to its simplicity, user-friendly interface, and open-source nature. Within Taguette, interviewees’ responses were systematically tagged using predefined thematic categories: *Alternatives or importance of products, Corruption and legal controls, Definition of practices, Price information, Sources of products, Uses of products,* and *Other interesting information.* These categories were decided based on a review of the bibliography, discussions during the initial research phase with authorities and NGOs, as well as insights from other similar studies. They were then refined after the interviews during the analyses phase. We also used these interviews to build a typology of users of felid parts. We categorized individuals based on their self-identified roles and the unique patterns of use they described [44]. The actual sentences used by interviewees are presented in *italics* and in quotes to illustrate their points. These quotes were translated from French into English, except when a keyword or concept could not be translated, in which case the local names or expressions were kept [45]. 

To identify the possible roads facilitating product movement across West Africa, we systematically collected information on each product’s reported origin and destination from interviewees. Each potential itinerary was mapped from its starting point (Point A) to its endpoint (Point B) using the latest version of Google Maps (online version: “https://www.google.com/maps (accessed on 3 October 2024)”) to determine the fastest route between the two points. We then pooled all mapped itineraries together to create a comprehensive dataset of potential terrestrial transportation pathways. The heatmap was generated using QGIS version 3.20 (Odense) [46], by using the “density of lines” function, with the density of lines reflecting likely route usage. The more lines on a segment, the more prominently it appeared on the heatmap. This approach assumes that decision-making by wildlife traffickers on route selection is primarily driven by minimizing travel costs or maximizing speed to destination over minimizing risks of detection and apprehension. This assumption was made after an interview with a seller explaining his itinerary (ID42) and another who stated that his supplier was using public transportation along the fastest routes (ID82). 

## 3. Results

The surveys were conducted from April 17 to June 17 across 16 cities and villages in Côte d’Ivoire. In total, 46 markets were inspected, of which 29 contained visible wildlife products. A detailed assessment of the available wildlife products on offer was conducted across 93 stalls displaying these products. Additionally, 39 interviews were conducted with vendors at these markets and 14 were conducted with individuals who use wildlife products or engage in “mystique” practices (see Section 3.1 “Practitioner typology”), providing an in-depth perspective on the cultural practices and beliefs associated with these products within Côte d’Ivoire (see Figure 1).

Among the 16 cities visited, 13 had wildlife products available in their markets. Leopard-based products were found in 41 out of 93 stalls (approximately 42.7%), with the highest concentration in Bondoukou, followed by Yamoussoukro and Bouaké (Appendix B). Lion-derived products were present in 24 stalls (25%), with Bouaké showing the highest proportion, followed by Korhogo and Abidjan. African golden cat products were found in 9.7% of stalls, serval products in 23.7%, African wildcat products in 6.5%, and caracal products in 3.2% of stalls. Other species were also available: spotted hyenas (*Crocuta crocuta*) were found in 56 stalls (58.3%), highlighting their strong presence in the markets, while chimpanzees (*Pan troglodytes*) were found in only one stall, highlighting the rarity of certain species (Appendix B). Full lion skins were more expensive than those of leopards and spotted hyenas (Table 1).

Data on the origins and demographics of vendors, collected through interviews and informal discussions, indicated that the majority of sellers were men (87%), with estimated ages between 36 and 60 years (42%), primarily from Niger (30%), and predominantly belonging to the Hausa ethnic group (43%) (see Table 2 for further details).

### 3.1. Practitioner Typology

Our interviews allowed us to define what interviewees called the “mystique” as all practices deemed supernatural and distinct from religious rituals. It manifests in various activities such as initiation ceremonies, protective healing against misfortune, and the use of talismans (or “gris-gris”). These mystical practices are characterized by a strong syncretism, making them difficult to describe uniformly due to regional variations and the diversity of interpretations. Based on interviews with vendors and practitioners who use felid body parts, we detailed some terms that allow us to classify certain practitioners.

Marabout: The marabout described themselves as a spiritual leader whose practices are rooted in Islam. While some marabouts do not engage in mystical practices at all—activities that are disapproved of by more conservative branches of Islam like Wahhabism—many incorporate mystical elements into their services. This may include the use of talismans (or “gri-gri”), specific prayers, or mystical ablutions. As one interlocutor explained, “the difference [between marabout, “féticheur”, and traditional healer] is the use of the Quran” (ID6). They also highlighted a significant difference between their mystical practices and those of “féticheur”. The marabout primarily uses the Quran and prayers, which is perceived as “healthier” but also slower in its effects: “People go to them [the “féticheurs”] because this type of practice is faster than our prayers” (ID38).

“Féticheur”: In contrast to the marabouts, the “féticheurs” portrayed themselves as practitioners who do not rely on the Quran for their practices, even though they may be Muslim. Their mystical practices are highly varied but generally rely on the use of a fetish, a sacred object that serves as a link between the practitioner and the mystical world. One interviewee explained: “The “féticheur” adores his fetish; it is his working tool” (ID60). They explained that their practices of “féticheur” often require sacrifices, also referred to as “blood power.” These sacrifices aim to invoke or strengthen the powers of the spirits with which they work: “I use many domestic animals, like chickens, goats, and sheep. I make sacrifices” (ID40). These practices are perceived as very powerful and quick, explaining their popularity in certain communities. However, they can also carry negative connotations, particularly due to their association with bloody sacrifices. Some do not believe in the effectiveness of fetish practices and consider the power of Islam and God to be much stronger: “For me, practices with the Quran are much more powerful than sacrifices” (ID20).

Traditional healer: Traditional healers introduced themselves as holders of ancestral knowledge, primarily transmitted from generation to generation. They possess knowledge related to nature, especially medicinal plants, and use these to heal or alleviate ailments. Unlike the marabouts, who often incorporate elements of Islam into their practices, the tradition-al healer focuses on traditional remedies and direct connections with the spirits of nature. As one respondent explained: “The traditional healer is a holder of knowledge from our grandparents about remedies to heal. They know the plants, the prayers, and the methods to heal people. These are the traditional knowledge of our ancestors” (ID38).

Sorcerer: The term "sorcerer" was mentioned less frequently in our interviews, likely due to its pejorative connotation. Unlike the marabout or the “féticheur”, the sorcerer is often associated with secretive mystical practices, hidden from public view. These practices, generally conducted at night or subtly, are often linked to malevolent intentions or curses. One interlocutor clarified: “The difference with the sorcerer is that they [the sorcerers] do evil” (ID21).

Dozo: The dozos are members of a confraternity of hunters, initially stemming from the caste of hunters among the Manding ethnic groups, particularly the Bambaras. Over time, this practice has spread and democratized, transcending ethnic and geographical boundaries. Within the confraternity, dozos fulfill various roles, such as hunter, griot (i.e., traditional West African storyteller, historian and musician, who preserves and conveys oral history [47]) or traditional healer. Their knowledge and powers come from their initiation and experience in the bush, granting them a deep connection with nature and its mysteries. As noted in an interview: “The dozo is the one who seeks animals; they kill them out there and bring them home” (ID60). Another respondent added: “A dozo is not necessarily Muslim and does not practice with the Quran. Not everyone masters the “mystique” even among dozos, training differs” (ID20).

Each of these actors was reported to use skins or parts of animals in their practices, particularly those of lions and leopards. There is significant permeability among these different categories, often blurring the lines between them. Some marabouts, for example, may also identify as “féticheurs”, or some dozos adopt mystical practices similar to those of “féticheurs”. Likewise, traditional healers may incorporate animal products into their rituals, further blurring distinctions. This complexity renders generalization and typology challenging, as practices and beliefs are reported to vary considerably from person to person and region to region.

Practitioners also act as doctors, prescribing remedies to address physical or mystical ailments. As stated by ID47: "I prescribe the use of lion (or leopard) if it is needed for what the person coming to me is seeking." The user is then responsible for sourcing the prescribed products from sellers in the market or through their own means. In some cases, practitioners may also sell products themselves, such as skins or transformed items like “gris-gris”. For example, ID86 had a dedicated consultation room and a separate space for selling such items. Additionally, transformed products like “gris-gris” can be crafted by intermediaries, typically leatherworkers called “cordonniers”. In these cases, the user provides the raw materials and the practitioner’s prescription, and the artisan crafts the “gris-gris” accordingly.

### 3.2. Uses of Carnivore Products

The interviews revealed numerous traditional uses of animals (Table 3), particularly felids, with a diversity of associated practices and remedies. One of the most common uses is the prescription of "gris-gris"— small amulets that often include a piece of wild cat skin—by practitioners to protect the wearers. Each practitioner has their own methods, making it difficult to define a unique use for a specific species. Furthermore, different parts of the same species can have varied applications. The same part of an animal can also be prepared in different ways by the same practitioner to achieve different outcomes. As one traditional healer reported: “It’s not obvious to tell you what the lion and the leopard are for; it depends on why you come to see me. With all animals, we can make a remedy; it depends on the power you put into it” (ID19).

Animals are thus used as more or less powerful vectors of magical religious power, depending on the species they belong to. Certain specific virtues attributed to certain species stand out, often derived from the inherent characteristics of these animals. For example, the lion and the leopard are associated with symbols of strength and respect. One participant clarified: “The lion is for strength; the leopard too, but it’s for respect” (ID39). Mustelids are also used for strength, for example, the skin of the honey badger (Mellivora capensis) is used for protection against bullets and knives due to the perceived toughness of its skin. Another example is the use of products derived from the spotted hyena, reputed to bring luck in business or during travel, due to its ability to survive and find food: “The hyena, you see, doesn’t hunt, yet it always finds something to eat. […] If you want to succeed, you will use this [the skin of the hyena]” (ID56).

Another use of skins, particularly that of the leopard, is found in arts and crafts. These skins are often seen in entirely different markets targeting foreign tourists. During a visit to a craft market, we encountered a seller offering handbags made from leopard skin. After gaining his trust and with the introduction of a local guide, he showed us an entire skin used for making various products. Here, the esthetic value of the skin is sought after, similar to other skins deemed attractive, primarily those from reptiles (crocodiles, snakes) for the production of belts, handbags, and other leather goods. The seller also appeared accustomed to negotiating the sale of whole skins directly with foreigners.

Finally, local chiefdom (local kings) is also likely to purchase entire skins for prestige, status, decoration, and sometimes for ritual protection: “Only kings can use it [the entire skin of leopard]” (ID86).

### 3.3. Cultural Importance and Alternatives of Large Felids Skins

Our interviews showed that lion and leopard skins hold significant value in mystical and traditional practices, as they are believed to confer considerable power. Ensuring the quality and authenticity of these products is therefore essential, often serving as a key factor in establishing a seller’s reputation. As noted by one spiritual healer: “Among all of them, it’s only the leopard and lion that I use in my gris-gris” (ID86). However, due to the increasing scarcity of these large felines, alternatives are emerging, such as the use of smaller feline skins like the African golden cat. These smaller species can act as substitutes when lion skin is unavailable, though they are considered less potent: “Now that the bush is empty [of large felids] I replace them with other [feline skins], but it’s not the same, they have less strength!” (ID6). Some users prefer alternative approaches that avoid the use of feline skins altogether, opting instead for plants or specialized knowledge. For example, one practitioner explained: “I know a tree in the bush that represents the leopard and another for the lion, so I don’t need the skin—I can use the bark as a substitute” (ID20). 

### 3.4. Sources of Felid Products

Over 30% of respondents stated that wildlife products were acquired in Côte d’Ivoire, particularly leopard skins (Figure 2). More than half of the Ivorian respondents stated that these products were procured from northern towns, such as Korhogo, reflecting a marked flow of wildlife products from these areas to major southern cities, notably Abidjan. This trade pattern was reinforced by multiple testimonies, highlighting Korhogo’s central role in this trade, and our analyses of the itineraries (Figure 3). As one respondent said: “It often comes from Korhogo” (ID18). Additionally, the Taï and Comoé National Parks are each mentioned once as origins of leopard skins. One interviewee remarked: “Before, I used to go to Taï for leopard skins, but now it’s not as easy to get them” (ID41). Aside from these parks, the exact origins of other specific felid populations from which the products were extracted could not be resolved from interviews. In some cases, such as Mali, the country of felid product procurement described by interviewees often did not possess any extant populations of leopard or lion, suggesting these were transit countries. The trade pattern resolved here therefore primarily reflects the immediate connections between Côte d’Ivoire and supplying, rather than origin, countries.

Neighboring countries, particularly Sahelian countries like Mali, Burkina Faso, and Niger, were also frequently cited as points of procurement for vendors of products derived from large felines. Nigeria serves not only as a key transit point but also as an origin point for many wildlife vendors and products. One respondent indicated: “My brothers from Nigeria bring these [feline skins]” (ID17), underscoring the connection between product origins and Hausa trade networks operating between Nigeria and Côte d’Ivoire. Other West African countries, such as Senegal, Guinea, and Ghana, were also mentioned as potential points of procurement of leopard and lion products, although these references were less frequent. 

Finally, countries from Central Africa (Gabon and Cameroon), East Africa (Kenya), and Southern Africa (Zambia and South Africa) were also mentioned and should not be overlooked given their considerably larger feline populations compared to West Africa. As one respondent stated: “The leopards come from South Africa” (ID50). 

The analysis of itineraries suggests four main axes of interest. The northern Côte d’Ivoire city of Korhogo appears to be strategically placed at the intersection of trade routes to receive wildlife products from the Sahelian countries, particularly Mali to the northwest (*n* = 4 mentions), and Burkina Faso, and Niger to the northeast (*n* = 6). Within Côte d’Ivoire a prominent and well-supported (*n* = 7) north-south axis then links Korhogo to markets in Bouaké, Yamoussoukro and Abidjan along the A3 highway. A prominent coastal route then connects the capitals of neighboring coastal countries, Ghana, Benin, Togo, and Nigeria to Abidjan’s markets (*n* = 7) (Figure 3).

### 3.5. Enforcement and Corruption

The ban on selling lion and leopard skins in West Africa was widely recognized by the interviewees, as evidenced by numerous testimonies from sellers. This ban was mentioned in 13 different interviews (34%). Sellers acknowledged that these animals are protected by law and that possessing or selling their parts is illegal in Côte d’Ivoire. For instance, one seller stated: “Everyone knows that it’s prohibited” (ID5), while another confirmed: "No, I no longer sell prohibited animals" regarding lions (ID6).

Concurrently, corruption was also cited as a prevalent factor by the interviewees, often used to circumvent legal controls by both sellers and traffickers. Corruption was mentioned in 10 different interviews (26%). Sellers indicated that illegal payments to law enforcement officers sometimes allowed them to bypass bans: “The authorities hassle us with that, but if you pay, you don’t have any problems and can continue selling” (ID29). Another seller explained: “We paid a fee to be here… every month, we pay a little” (ID53). Additionally, some users reported that traffickers can pay at borders to avoid sanctions: "Often, you can get arrested, but you just need to pay to get through" (ID56).

## 4. Discussion

This study represents the first investigation into the illegal trade of felid products in Côte d’Ivoire, highlighting its deeply ingrained nature as a practice that transcends societal and geographic boundaries. While the scale of the trade requires further exploration, such practices could have particularly severe consequences in regions where felid populations, such as those of lions and leopards, are already in precarious states [48]. We identified key practitioners using the skins and other parts of those felids in their practices and highlighted the diverse cultural uses of different felid parts. Additionally, we provided insights into illegal wildlife trade routes and sources of products, as well as enforcement gaps and potential corruption of law enforcement officers.

During our itinerary (Figure 1), lion and leopard parts were found in 85% of the cities investigated. This wide availability of felid products aligns with findings from similar studies conducted in Ghana, Niger, Benin, and Senegal [10,11,49]. Cities where no wildlife products were found were primarily located around Comoé National Park, where stronger enforcement by the local wildlife authority (l’Office ivoirien des parcs et reserves–OIPR) and the Ministère des Eaux et Forêts likely discourages the establishment of permanent wildlife product stalls. Another possible explanation is that these smaller cities rely on itinerant vendors who sell wildlife products sporadically, making their presence less consistent during our surveys [49].

The majority of sellers identified in our study were Hausa men, often originating from Niger and Nigeria, confirming their dominance in regional trade networks, as established by prior studies on illegal wildlife trade of felids products [10,11]. This could be explained by the Hausa’s historical role as a trade-oriented and nomadic herder community, with extensive experience and networks in commerce and mobility across West Africa, enabling them to play a central role in these markets [50]. The Senoufo were the second most frequent sellers, a finding that complements existing literature emphasizing their cultural proximity to the Dozo confraternity and their integration into the Manding world, where the use of felid products is deeply embedded [41]. Sellers from southern ethnic groups, predominantly Christian communities, were less commonly observed, although evidence collected here and from other sources suggests that they may also participate in the trade as users or sellers [10]. However, the study’s limited sampling, particularly its lack of spatial coverage within Côte d’Ivoire, may have impacted the representativeness of the observed ethnic distribution, requiring caution when interpreting these findings. For further study on the trade, we recommend expanding the scope and designing the study statistically to provide more comprehensive data on the scale of the trade in Côte d’Ivoire. Additionally, we suggest conducting long-term tracking and interviews with volunteer sellers, carried out by local and Hausa-speaking researchers. Such interviews could help bypass some of the biases inherent in this study and improve knowledge of trade routes, frequency, and seller profiles. However, due to the illegal nature of this trade, building trust—even with native speakers—will remain a significant challenge for such studies.

Practitioners identified in various ways and reported a wide variety of uses for felid products allowing us to define a typology of practitioners. This characterization, though highly plastic and permeable, provides a clearer understanding of certain groups, which would be useful in audience delineation for targeted behavior change campaigns [36]. 

However, due to time constraints, our study focused solely on practitioners, excluding end-users, whose profiles and practices remain understudied. A limitation of our study was that we did not include specific questions that would allow for a better understanding of the demographics and motivations of end-users. As one practitioner noted, “it can be anyone—women, men, Christian, Muslim—I heal everyone” (ID53). This highlights the diversity of individuals seeking these services. However, some uses of felids, such as for wealth, strength, or protection, appear more targeted toward specific groups, such as politicians, the military, or those in dangerous professions. For instance, another practitioner stated, “politicians, military—they come to me to find protection” (ID82), a pattern also observed in Ghana and ongoing studies in Senegal [10,49]. We recommend conducting a large-scale end-user survey, designed as a consumer survey, to gain deeper insight into the profiles of end-users and better inform targeted behavioral change campaigns. Such a survey would also help determine whether practitioners are driving the demand or if it is primarily consumer-led.

Individuals from various backgrounds may seek practitioners using wild felids for a wide range of purposes, especially for mystical protection that does not exist in conventional medicine. This may represent the most deeply embedded cultural reason for such uses. Another key factor could be the cost and availability of practitioners. While our study did not systematically collect data on the costs of practitioners’ services, we observed a wide range of prices, from a few hundred CFA (approx. USD 1) (ID52) to 300,000 CFA (approx. USD 500) for an anti-bullet belt (ID83). We argue that practitioners’ high availability, combined with their adaptive pricing, makes them a more competitive solution compared to Western medicine, which is much more expensive, and often more difficult to access due to limited proximity of pharmacies or unavailability of certain medicines [51]. A more comprehensive and representative sampling of end-users would be critical to inform effective interventions. Additionally, our investigation identified the presence of felid products in craft markets and their use in local chiefdoms, practices also documented in the literature [10,15,48]. However, these environments were not directly investigated in this study, and we recommend conducting targeted research in these contexts.

Our study revealed the significant cultural importance of felid products in Côte d’Ivoire, which are associated with a wide range of uses. For groups like the Dozos, who are deeply connected to nature and the bush and hold a respected and organized social structure, these products carry both symbolic and functional values [52]. This connection suggests that the Dozos could be key partners in behavior-change campaigns aiming at limiting the use of felid products, as their influence on Ivorian society might help facilitate shifts in traditional practices. Dozos could serve as local champions, adopting a new social role as protectors of wild felids and advocates for raising awareness of end-users and other practitioners within the frame of behavior-change campaigns [53]. However, further research is needed to evaluate the feasibility of this role and ensure it would align with their identity as hunters, rather than conflicts with it.

Despite the significant cultural importance of large-felid products, the plasticity of these practices suggests that some modifications are possible. For example, declining lion populations may explain the higher prices observed for lion skins (Table 1) and could drive a shift toward the use of smaller felid species [54]. Additionally, nine (about 10%) interviewees mentioned alternative products, such as certain tree species, as substitutes for felid products. This indicates the potential for preserving cultural practices while reducing reliance on endangered species. However, the tree species in question was not identified, raising concerns about its potential conservation status. Collaborating with cultural leaders would allow us to identify and promote acceptable symbolic alternatives to felid skins, ensuring cultural traditions are respected while protecting wildlife. Follow-up research could explore other viable alternatives beyond those identified here. 

Our interviews revealed that corruption exists and plays a role in enabling wildlife trafficking, undermining enforcement efforts. During one of our discussions, we asked a seller denouncing corruption what he thought about it, and he stated, “What can we do? This is the game; we have to pay if we want to sell our products” (ID57), reflecting a certain acceptance and normalization of corruption. Côte d’Ivoire ranks 87th out of 180 countries on Transparency International’s Corruption Perception Index (https://www.transparency.org/ (accessed on 22 October 2024)”, an improvement of three places from the previous ranking, though ongoing challenges remain. Several initiatives, including treaties under CITES and capacity-building programs such as USAID’s WABiLED initiative, demonstrate progress in addressing these issues [55]. To further combat corruption, we recommend implementing the ECOWAS R12 directive, which calls for a regional strategy to tackle wildlife crime by identifying priorities and harmonizing national and regional efforts [56]. Strengthening controls, such as increasing checkpoints along key routes like the Korhogo-Abidjan corridor and borders with Burkina Faso, is crucial. Most importantly, enhancing training for enforcement agencies is essential, as we noticed a lack of awareness among law enforcement personnel about the importance of protecting wild cat species, during informal discussions. References suggest that training programs can effectively sensitize individuals to the significance of wildlife conservation and reduce their susceptibility to corruption [57]. However, these efforts must be paired with governance reforms aimed at reducing corruption

The most reported source countries for this trade were Sahelian nations, confirmed by other studies and data from the Illegal Wildlife Trade Portal (Appendix C) [10,11,49]. However, these countries often represent the origin of traders rather than the origin of felid populations from which derived products are made. We recommend developing a regional database of lion and leopard skin samples for genetic analysis to identify the geographic origins of trafficked items. The Sahelian corridor plays a key role in the trade of felid products, as evidenced by our findings that identify Mali, Niger, and Burkina Faso as major points of procurement of these products, particularly according to the vendors we interviewed. Many of the vendors are Hausa, often linked to nomadic pastoralism or directly involved in pastoralist communities, and who originate from Niger or Nigeria [53]. Our road network analysis suggests that Sahelian trade routes are likely pathways for transporting these goods. This corridor, already recognized for facilitating the trafficking of arms, people, and gold, further highlights its broader involvement in illicit activities, including the trade of wildlife products [54]. Our analysis connects these observations to the broader context of regional trade routes, aligning with the evidence we gathered during the study. Nigeria, frequently described as a strategic commercial hub connecting West Africa to the broader continent [1,58], also plays a pivotal role in this network. The frequent seizures of wildlife products in Nigeria further validate its central role in these illegal trade networks [11,49,59]. Additionally, a coastal route from Abidjan to other West African nations, including Ghana, Benin, Togo, and Nigeria, supports intra-regional trade and reinforces economic connectivity along the West African coast. These routes are consistent with prior research documenting wildlife product trafficking across the region [10,11,60]. Despite the insights provided by our heatmap, this analysis remains exploratory and carries certain limitations. Google Maps was used to estimate the fastest routes, but traders may choose alternative paths to avoid checkpoints or to incorporate additional stops. However, the limited road network in Côte d’Ivoire, coupled with potential corruption that facilitates smoother passage even on the main roads, likely minimizes these deviations, lending credibility to the identified routes. Another limitation of this technique is that we used the capital of each neighboring country as the starting point for the route itinerary when specific locations were not provided. In reality, traffickers may operate from a wide range of locations within these countries. Nevertheless, we consider these routes as valuable findings because they highlight the proximity of trade routes to key areas of felid presence. For instance, the routes’ proximity to lion populations in Senegal and leopard populations in neighboring countries suggests these areas could be potential sources. Additionally, the W-Arly-Pendjari (WAP) complex, which hosts the largest lion population in West Africa, is particularly vulnerable and may be targeted to supply wildlife products for trade in Côte d’Ivoire. Finally, we documented records of traded products potentially sourced beyond West Africa, such as in Gabon and Kenya. These extra-West African countries also appear in seizure data from the Illegal Wildlife Trade Portal (Appendix B and C), underscoring the scale of the trafficking and the potential threat to felid populations in these regions. Furthermore, seizure records of exports from Côte d’Ivoire reveal connections to Europe and Asia, particularly China and Vietnam (Appendix C). This suggests the existence of other potential illegal trade routes, possibly connected to other products like ivory, bushmeat, or pangolin scales, further emphasizing the need for broader investigations into the dynamics of wildlife trafficking networks.

## 5. Conclusions

This study provides the first insight into the trade of wild felids in Côte d’Ivoire, highlighting its cultural, medicinal, and economic dimensions. Through interviews with vendors and practitioners, we identified trade routes connecting West African countries and instances where corruption facilitates trade. While the study is limited in scale, it reveals consistent patterns of use and trade, emphasizing the need for further research to better understand consumer motivations, supply chains, and the socio-economic drivers of this trade. We recommend expanding the scope of future studies to include end-user surveys designed to uncover the profiles and behaviors of consumers, as well as long-term tracking and interviews conducted by local and Hausa-speaking researchers to reduce biases and enhance knowledge of trade routes, frequency of use, and seller profiles. Additionally, strengthening enforcement training, implementing regional strategies like the ECOWAS R12 directive, and pairing these efforts with governance reforms to tackle corruption are crucial steps to combat this illegal trade effectively.

## Figures and Tables

**Figure 1 animals-15-00451-f001:**
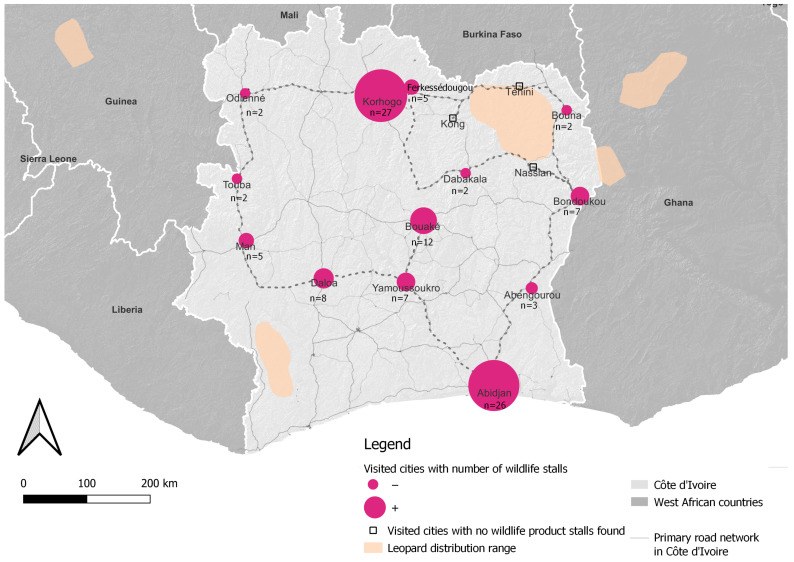
Map showing the cities investigated during market surveys in Côte d’Ivoire. Market stalls selling wildlife products (pink circles, size indicating the number of stalls, *n* = X) and extant and possibly extant leopard distribution zones (light orange) in Côte d’Ivoire and neighboring countries.

**Figure 2 animals-15-00451-f002:**
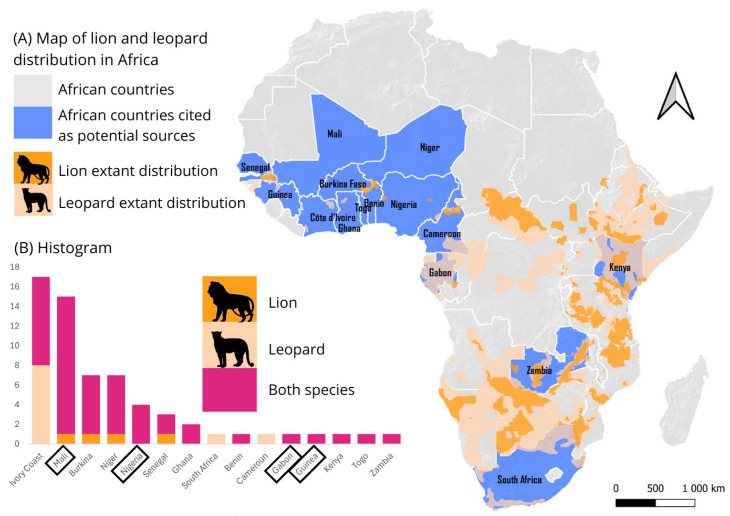
(**A**) Map of Africa showing lion and leopard distributions according to the latest IUCN Red List distribution maps (orange shades), frequency of countries cited by the respondents as potential points of procurement of felid products, or seller origins (blue). (**B**) Histogram of the number of times a country was cited in the interviews as the origin of products or sellers, according to the species (lion in orange, leopard in beige, both species in pink). Countries encased in black are corroborated by seizure data from the Illegal Wildlife Trade Portal.

**Figure 3 animals-15-00451-f003:**
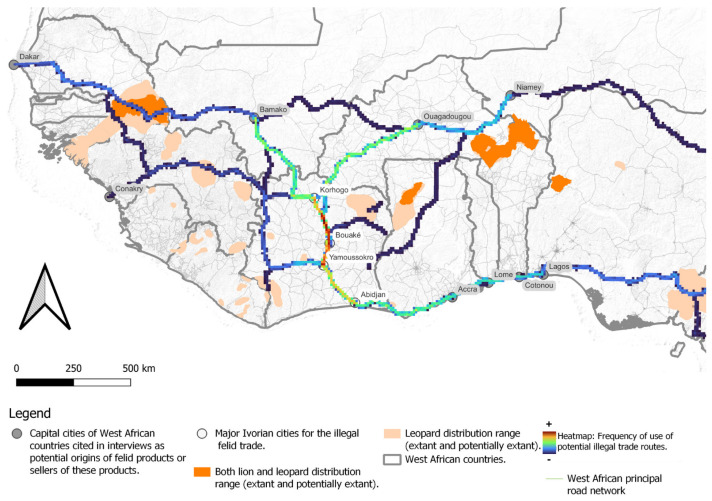
Heatmap of potential trade routes used in the illegal wildlife trafficking of felids to Côte d’Ivoire and distribution of remaining lion and leopard populations in West Africa, according to the latest IUCN Red List distribution maps.

**Table 1 animals-15-00451-t001:** Mean and range of reported prices for large carnivore products, as cited by 17 individuals, during interviews and informal discussions conducted with sellers and practitioners in Côte d’Ivoire between April and May 2024.

Wildlife Product	Frequency Cited	Average Price XOF (Approx. $)	Price Range XOF (Approx. $)
Lion skin	6	950,000 (approx. $1550)	600,000–1,500,000 (approx. $980–$2450)
Leopard skin	7	305,000 (approx. $500)	200,000–500,000 (approx. $330–$820)
Lion tail	1	75,000 (approx. $120)	N/A (not available)
Lion claw	1	4000 (approx. $6.50)	N/A
Lion fat (small piece)	1	1000 (approx. $1.60)	N/A
Spotted hyena skin (whole)	1	300,000 (approx. $490)	N/A

**Table 2 animals-15-00451-t002:** Summary of seller demographics and origins in the 46 studied markets across Côte d’Ivoire.

Category	Sub-Category	Proportion (%)
Gender of Sellers	Men	87%
	Women	7.50%
	Not specified	5.40%
Estimated age of Sellers	18–35 years	18%
	35–60 years	42%
	Over 60 years	20%
	Not specified	19%
Country of Origin	Niger	30%
	Côte d’Ivoire	21.50%
	Burkina Faso	11.80%
	Nigeria	7.50%
	Mali	5.40%
	Benin, Ghana, Senegal (each)	1%
	Not specified	21.50%
Ethnic Group	Hausas	43%
	Senoufos	14%
	Mossis	10%
	Agnis, Ashantis, Bambaras, Djiminis, Igbos, Fulani, Wolofs (each)	1–2%
	Not specified	24%

**Table 3 animals-15-00451-t003:** Different uses of various large carnivore body parts used in traditional practices and quotes from interviewees highlighting those uses.

Parts Used	Uses	Quotes
Skin, fat (lion, leopard)	Power and respect	“*It has the therapeutic virtue of making a man tough, that is, influential and imposing, so that no one dares to contradict you.*” (ID48—Traditional healer)
Skin, fat (lion, leopard)	Mystical power	“*If someone wants to harm you, I use the lion to prevent harm (it’s a form of protection).*” (ID42—Traditional healer)
Skin, fat (lion, leopard)	Royal power	“*This ointment with lion and leopard skins is used by kings to inspire respect and authority.*” (ID86—Féticheur)
Skin, fat (lion, leopard)	Political and military power, respect	“*Presidents and military personnel use it a lot to inspire fear and respect. There are also plants that can do the same.*” (ID96—Dozo)
Skin (leopard)	Witch detection	“*The leopard will come, and the witch will be forced to leave (it drives away witches).*” (ID42—Traditional healer)
Skin (lion)	Wealth	“*You can divert money with this, up to 20 million XOF with the lion’s skin.*” (ID41—Marabout)
Skin (lion)	Couple separation	“*I make gri-gri with lion skin to separate couples.*” (ID85—Dozo)
Fat (lion)	Therapeutic care	“*For fractures and back pain, lion fat is used.*” (ID96—Dozo)
Skin (golden cat)	Wealth	“*It gives you money.*” (ID82—Marabout)
Skin (serval)	Amulet crafting	“*I use this one as leather to make the gri-gri, it lasts very long*” (ID96—Dozo)
Skin (small cats)	Substitute	“*If you can’t find [leopard or lion skin] or you don’t have the money for it you can use this one’s*” (ID96—Dozo)
Skin (spotted hyena)	Success	“*The hyena doesn’t hunt, yet it always finds something to eat. […] If you want to succeed, you will use this [the skin of the hyena].*” (ID56—Traditional healer)

## Data Availability

The data presented in this study are available on request from the corresponding author due to privacy and ethical reasons.

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
