# Peer review of "Fading Roars? A Survey of the Cultural Use and Illegal Trade in Wild Felid Body Parts in Côte d’Ivoire"

_animals, 2025, doi:10.3390/ani15030451_

Round 1

Reviewer 1 Report

Comments and Suggestions for Authors

This article concerns trade in wild felids in West Africa, and focuses on sellers of felid products, as well as transport routes of products. My biggest comment is that, while this study is very well-written and seems to be robustly researched, not much is made of the data. It looks like there will have been a lot of data based on the number of interviews and different objectives, but this isn’t represented within the manuscript. I was super excited by the questions on the interview guide given in Appendix A, and disappointed by how little information was shared. I also realized, once I was reading the Discussion, that the Methods was missing description of qualitative analysis (e.g. thematic analysis?), because it looks as though that kind of analysis was not done here. I strongly suggest doing such analysis as it will greatly increase the resonance of this study. For guidance, I suggest the authors consult other qualitative articles on IWT - there are a number of excellent studies that have come out of Africa and could serve as good models for this article (see for example Titeca, 2019). If the authors are unfamiliar with how to do qualitative analysis, I would suggest reading H Russell Bernard’s Research methods in anthropology: Qualitative and quantitative approaches. I would also suggest checking out a program such as NVivo, which greatly helps with thematic analysis, particularly for researchers who are new to this type of analysis.

Titeca, K., 2019. Illegal ivory trade as transnational organized crime? An empirical study into ivory traders in Uganda. The British Journal of Criminology, 59(1), pp.24-44.

Other than this, my comments are minor. I have included them below, along with more specific comments intended to help the authors with making use of their data and increasing the impact of this work.

Introduction

Line 54: Which region?

Lines 75 – 77: This sentence should be moved higher in the manuscript to the discussion about trafficking. The sentence following this refers specifically to use of felid parts for traditional uses, which flows better with the preceding sentences.

Line 128: I suggest removing this sentence, since it is a result.

Materials and Methods

Overall the Methods are good but should be separated into sub-headings (e.g. “Study Area”) for more clarity. Also, when discussing the interview guide(s?), it will help the reader orient to include brief description of what questions were asked, and why.

Line 173: Did you have these questions prepared in advance? I think it would be helpful to include as an Appendix.

Results

Line 222: Which year?

Table 1, Lines 230 – 240: It’s worth clarifying, for full transparency, whether only 17 individuals cited felid products’ use, or whether only 17 individuals mentioned price.

Line 245: “Ages”

Discussion

Line 417: I’m not convinced that your results show that this is widespread. I think, however, that you can reframe this assertion to comments more specific to what you found. For example, felid trade as an ingrained practice that transcends across societies and borders, and any level of trade having an impact on leopard and lion viability.

Lines 417 – 421: This section highlights for me how much more information could be gleaned from your data, which isn’t presented here (see my broader comments about this). For example, you note that you identified corruption as existing, but is there anything from your data that could help practitioners to understand why this corruption occurs, what the level of burden to pay is compared to profit, or what possible actions could be taken to address this issue? Everything you present in the Results, while interesting, is quite high level and wouldn’t be super helpful for informing targeted behavior change efforts or certain policy changes. I suggest returning to your data and “going deeper” into what you gathered, to make those kinds of connections that can lead to more impactful insights.

Line 422: This figure and accompanying description should be in the Results.

Line 424: Include missing references 😊.

Line 427: Reference?

Line 436: It would be helpful to include a brief comment as to why this may be the case (a more mercantile-inclined ethnic group?).

Line 441: Is this evidence you collected? Or is this from other sources?

Lines 445 – 448: While this is true, I again think that section is missing some insights. Crucially, I think it would be helpful to provide more detail on who uses each group’s services. This would help with consumer-focused behavior change campaigns, versus those focused on the “suppliers” (these different groups).

Lines 449 – 450: Per my above point, it’s true that you couldn’t get demographic information about these individuals, but I suspect you can still gather insights into who was using these services. If not, that should be clarified here (e.g. “a limitation of our study was that we did not include questions that would allow for understanding of users of the practitioners’”). I also think this could be the place for more thoughtful reflection on the motivations you did identify, such as to bring wealth, or for medicinal purposes. You can note that your research is important for identifying these specific motivations and you can highlight possible messages for behavior change efforts.

Lines 470: How many?

Lines 472 – 476: It’s also worth noting that there is potential for follow-up research to directly explore other alternatives beyond that identified.

Line 477: At present, you don’t provide evidence that corruption is a significant factor. If you wish to make that argument, a good first step would be to explicitly state how many of your interviewees mentioned corruption, and provide more quotes showing prevalence (and, if present in the interviews, acceptability of corruption).

Line 487: I would think also some “easy” first steps such as increasing the pay of enforcement officers? It would be good to discuss here (even or perhaps especially if it is infeasible).

Lines 495 – 496: This is a bit random, and as far as I can tell doesn’t appear in your results. Did you collect data on how products are transferred in the region? It looks like you didn’t, but if you did that should be included in the Results, if you wish to discuss it here.

Lines 530 – 533: Again, I am not convinced your study shows this. This study is an exploratory qualitative study, which certainly has significant value, but can’t be used to make such assertions, particularly as you did not investigate consumers and by extension, the prevalence of actual consumption. I think you could perhaps make this argument more convincing if connected to your earlier comments around the prevalence of vendors selling felid products, along with some quotes/insights from the vendors/practitioners around the availability of products and perceived level of demand. I also think you need to include more direct connection to population viability – has anyone analyzed the effect of threats on felid populations? E.g., models assessing threat impact? If so, that would be necessary to include here, so that you could make those assertions about trade being “severe”.

Lines 533 – 534: I would think, per your findings around corruption, that your findings don’t actually highlight the importance of enforcement. I suggest removing this/rewording to focus only on monitoring and how it could be helpful.

Lines 535 – 537: This is great and true, but a missing piece of this article is your suggestion of how that could be accomplished, based on your data. This is where thematic analysis can play a really crucial role.

Author Response

General comment

This article concerns trade in wild felids in West Africa, and focuses on sellers of felid products, as well as transport routes of products. My biggest comment is that, while this study is very well-written and seems to be robustly researched, not much is made of the data. It looks like there will have been a lot of data based on the number of interviews and different objectives, but this isn’t represented within the manuscript. I was super excited by the questions on the interview guide given in Appendix A, and disappointed by how little information was shared. I also realized, once I was reading the Discussion, that the Methods was missing description of qualitative analysis (e.g. thematic analysis?), because it looks as though that kind of analysis was not done here. I strongly suggest doing such analysis as it will greatly increase the resonance of this study. For guidance, I suggest the authors consult other qualitative articles on IWT - there are a number of excellent studies that have come out of Africa and could serve as good models for this article (see for example Titeca, 2019). If the authors are unfamiliar with how to do qualitative analysis, I would suggest reading H Russell Bernard’s Research methods in anthropology: Qualitative and quantitative approaches. I would also suggest checking out a program such as NVivo, which greatly helps with thematic analysis, particularly for researchers who are new to this type of analysis.

Titeca, K., 2019. Illegal ivory trade as transnational organized crime? An empirical study into ivory traders in Uganda. The British Journal of Criminology, 59(1), pp.24-44.

Other than this, my comments are minor. I have included them below, along with more specific comments intended to help the authors with making use of their data and increasing the impact of this work.
We thank the reviewer for the time and effort put into providing thoughtful and constructive comments that we think improved our paper.
We have modified the Methods section to address the reviewer’s concerns regarding the number of interviews and the scope of questions provided in Appendix A. Specifically, we clarified that this was a short, preliminary study conducted over two months across the whole country (including transport time to travel from one place to the other), and interviews with vendors were typically brief, meaning that not all questions were consistently asked, as mentioned in the interview guide.
Additionally, we acknowledge an omission in our Methods: we had not mentioned the use of the qualitative analysis software “Taguette” in transcribing interviews and tagging responses under several predefined themes. We have now added a paragraph on this tool and how we used it in the manuscript. While this method does not reach the complexity of advanced thematic analysis, it provided an accessible and structured way to analyze this exploratory dataset. We emphasize that this as a foundational first step into a topic that has not yet been addressed in the literature in Côte d’Ivoire.
We greatly appreciate the reviewer’s recommendation to consult Titeca, K. (2019). We acknowledge the high quality and depth of that work, which resulted from more than five years of detailed field investigation. We could not achieve the same level of precision and depth over two months of fieldwork with no previous study conducted on the topic in the country. However, we have learned from this recommendation and attempted to enhance the framing of our study accordingly.
In summary, we have adjusted the Methods to temper expectations regarding the study’s scale and scope. Additionally, we have refined the Conclusion to position this research as a broad overview and a crucial first insight into the trade in felids in Côte d’Ivoire. We emphasize that this study highlights the need for more in-depth investigations to inform targeted behavior change campaigns and other interventions to address the issue effectively.

Introduction

Line 54: Which region?
We replaced “that region” by West Africa to be more precise, thank you.

Lines 75 – 77: This sentence should be moved higher in the manuscript to the discussion about trafficking. The sentence following this refers specifically to use of felid parts for traditional uses, which flows better with the preceding sentences.
We moved the sentence accordingly. L.61-63

Line 128: I suggest removing this sentence, since it is a result.
Thank you for pointing that out, we removed the sentence.  

Materials and Methods

Overall the Methods are good but should be separated into sub-headings (e.g. “Study Area”) for more clarity. Also, when discussing the interview guide(s?), it will help the reader orient to include brief description of what questions were asked, and why.
Thank you for your suggestion. We have added sub-headings in the Methods section to improve clarity: Study area and market identification methods, Local guides and interview facilitation, Observations and interviews in wildlife markets, Interviews of practitioners, Ethical considerations and participant consent, Data reliability and analyses Additionally, we have included a brief description of the most frequently asked questions and clarified that not all questions were asked in every interview due to time constraints and the willingness of vendors to take time of their activity to respond to us.

Line 173: Did you have these questions prepared in advance? I think it would be helpful to include as an Appendix.
Thank you for pointing this out. The questions were already included in Appendix A, but we had not referenced it in the text. This has now been corrected to ensure clarity.

Results

Line 222: Which year?
We have now added the year (2024) to clarify this detail.

Table 1, Lines 230 – 240: It’s worth clarifying, for full transparency, whether only 17 individuals cited felid products’ use, or whether only 17 individuals mentioned price.
We have clarified this point by modifying the title of the figure to ensure full transparency.New title is “Mean and range of reported prices for large carnivore products, as cited by 17 individuals, during interviews and informal discussions conducted with sellers and practitioners in Côte d’Ivoire between April and May 2024.”

Line 245: “Ages”
Thank you for pointing this out. We have corrected the typo and modified "Ages" accordingly.

Discussion

Line 417: I’m not convinced that your results show that this is widespread. I think, however, that you can reframe this assertion to comments more specific to what you found. For example, felid trade as an ingrained practice that transcends across societies and borders, and any level of trade having an impact on leopard and lion viability.
Thank you for your feedback. We have rephrased the paragraph to focus on the insights specific to cultural use of the felid trade, reframing the assertion as suggested. We also added a reference proving that cultural use has an impact on felids viability in the case of lions. While we agree that the scope of our study may not be large enough to definitively demonstrate that the trade is widespread, we believe that it highlights a consistent pattern of felid use and trade across our study area, encompassing various uses and actors. This pattern aligns with findings from other countries (e.g., Ghana, Senegal, Benin, Niger, Burkina Faso).

Lines 417 – 421: This section highlights for me how much more information could be gleaned from your data, which isn’t presented here (see my broader comments about this). For example, you note that you identified corruption as existing, but is there anything from your data that could help practitioners to understand why this corruption occurs, what the level of burden to pay is compared to profit, or what possible actions could be taken to address this issue? Everything you present in the Results, while interesting, is quite high level and wouldn’t be super helpful for informing targeted behavior change efforts or certain policy changes. I suggest returning to your data and “going deeper” into what you gathered, to make those kinds of connections that can lead to more impactful insights.
Thank you for your feedback. As mentioned in our response to the general comment, this study serves as a first insight and monitoring of the felid trade situation in Côte d'Ivoire. We have made several modifications in an attempt to address the reviewer’s comment and the general feedback, but we also need to acknowledge that our data is not yet sufficient to go deeper into many aspects, including the reasons for corruption (as detailed in our response below about corruption).
We argue that the study still provides an interesting broad picture that can guide more in-depth monitoring and future studies aimed at addressing behavioral change campaigns. We have also rephrased the conclusion accordingly (as noted in our response below).

Line 422: This figure and accompanying description should be in the Results.
We have moved the figure and its accompanying description to the Results section as suggested.

Line 424: Include missing references ?.
Missing references included.

Line 427: Reference?
Missing reference included.

Line 436: It would be helpful to include a brief comment as to why this may be the case (a more mercantile-inclined ethnic group?).
Thank you for the suggestion. We have included a brief comment to explain this observation and added a relevant reference to provide further context for readers. L.477-479

Line 441: Is this evidence you collected? Or is this from other sources?
We have clarified that this evidence comes from both our own data collection and other sources. Additionally, we have added the appropriate references for the external sources.

Lines 445 – 448: While this is true, I again think that section is missing some insights. Crucially, I think it would be helpful to provide more detail on who uses each group’s services. This would help with consumer-focused behavior change campaigns, versus those focused on the “suppliers” (these different groups).
Please see our response to the comment just below as both comments are linked.

Lines 449 – 450: Per my above point, it’s true that you couldn’t get demographic information about these individuals, but I suspect you can still gather insights into who was using these services. If not, that should be clarified here (e.g. “a limitation of our study was that we did not include questions that would allow for understanding of users of the practitioners’”). I also think this could be the place for more thoughtful reflection on the motivations you did identify, such as to bring wealth, or for medicinal purposes. You can note that your research is important for identifying these specific motivations and you can highlight possible messages for behavior change efforts.
Thank you for your detailed and constructive feedback. We fully agree with both points raised. In response, we have clarified in the text the limitation of our study in not focusing on end-users. We chose to prioritize practitioners for several reasons: practitioners act as prescribers, essentially driving the use of felid products, making them a critical entry point for behavioral change campaigns. They are also more accessible for discussion compared to sellers, as they are not directly involved in illegal trade, and were easier to reach within the limited timeframe of this study. A broader end-user survey would require a more extensive consumer study, which was beyond the scope of this short-term project.
We revisited our data and added substantial new content to address your comments. Specifically, we noted that while practitioners’ services are accessible to anyone, certain types of felid products appear tailored to specific professions, particularly politicians and police or military persons who seek specific forms of protection. We also highlighted that this type of practice often addresses mystical illnesses or issues and is available across a wide range of prices. This adaptability aligns well with the local cultural context and provides an economically accessible alternative to western medicine.

Lines 470: How many?
We have added the missing number along with the corresponding percentage for clarity.

Lines 472 – 476: It’s also worth noting that there is potential for follow-up research to directly explore other alternatives beyond that identified.
This is true, we have added a sentence to acknowledge this possibility L.537-538.

Line 477: At present, you don’t provide evidence that corruption is a significant factor. If you wish to make that argument, a good first step would be to explicitly state how many of your interviewees mentioned corruption, and provide more quotes showing prevalence (and, if present in the interviews, acceptability of corruption).
Thank you for pointing this out. In response, we have removed the term "significant" and added the number of mentions of corruption in the Results section for greater transparency L.450. We have also included a statement of acceptability from an interviewee, as we believe it demonstrates a certain normalization of the phenomenon L.539-541. As outlined in our interview guides (Appendix), corruption was not a primary focus of this study. However, spontaneous discussions with interviewees revealed at least 3 different types of corruption stated by the interviewees in the results section:
-Payments to sell illegal wildlife products inside markets (e.g., payment of a stall fee).
-Payments to cross borders with illegal wildlife products.
-Payments to law enforcement officers to bypass arrest or inspection.
While these findings emerged incidentally, we believe they are important and warrant highlighting and discussion in this study. They provide a foundation for understanding the mechanisms facilitating the illegal wildlife trade and should be further investigated in future research.

Line 487: I would think also some “easy” first steps such as increasing the pay of enforcement officers? It would be good to discuss here (even or perhaps especially if it is infeasible).
Thank you for your suggestion. We do not think that increasing the pay of law enforcement officers would be an effective or “easy” first step. Research has shown that salary increases alone do not automatically reduce systemic corruption, as corruption is often influenced by broader structural and cultural factors. For example, studies in Ghana and elsewhere have demonstrated that raising salaries without addressing accountability and oversight mechanisms may not significantly impact corruption and, in some cases, may exacerbate it (Foltz, 2015).
Instead, we agree with the ECOWAS recommendations cited in our bibliography, which emphasize the importance of targeted law enforcement training.  This recommendation is particularly important because, during our investigation, we observed that, as stated in section L.500, military and police officers are often themselves involved in the use of felid parts. Additionally, in discussions outside the scope of this study, we noticed a lack of awareness among law enforcement personnel about the importance of protecting wild cat species. Finally, we found references suggesting that training programs can effectively sensitize individuals to the significance of wildlife conservation and reduce their susceptibility to corruption. (Transparency International, 2010)

Foltz, J. D., & Opoku-Agyemang, K. A. (2015). Do higher salaries lower petty corruption? A policy experiment on West Africa’s highways. Unpublished Working Paper, University of Wisconsin-Madison and University of California, Berkeley, 12, 245.

Transparency International (2010). Anti-corruption and police reform. https://knowledgehub.transparency.org/assets/uploads/helpdesk/247_Anti_corruption_police_reform.pdf?utm

Lines 495 – 496: This is a bit random, and as far as I can tell doesn’t appear in your results. Did you collect data on how products are transferred in the region? It looks like you didn’t, but if you did that should be included in the Results, if you wish to discuss it here.
Thank you for pointing this out. We agree that the section may have seemed disconnected initially. However, in the Results, we provide significant evidence pointing to the Sahelian countries—Mali, Niger, and Burkina Faso—as the most cited sources of products, at least by the vendors interviewed. Furthermore, the majority of vendors we encountered are Hausa, originating from Niger or Nigeria. Our road network analysis also highlights Sahelian trade routes as potential pathways for these products.
In this section, we aimed to connect these findings with broader knowledge of the extensive trade routes within the Sahelian corridor, referencing other studies for support. To address your concern, we have revised this section to better align it with the results and make the connections more explicit and convincing. This should clarify how the data we collected supports the discussion presented here L.559-569.

Lines 530 – 533: Again, I am not convinced your study shows this. This study is an exploratory qualitative study, which certainly has significant value, but can’t be used to make such assertions, particularly as you did not investigate consumers and by extension, the prevalence of actual consumption. I think you could perhaps make this argument more convincing if connected to your earlier comments around the prevalence of vendors selling felid products, along with some quotes/insights from the vendors/practitioners around the availability of products and perceived level of demand. I also think you need to include more direct connection to population viability – has anyone analyzed the effect of threats on felid populations? E.g., models assessing threat impact? If so, that would be necessary to include here, so that you could make those assertions about trade being “severe”.
We agree with the reviewer’s points, as well as the two related comments below. In response, we have decided to completely rewrite the conclusion to provide a broader overview of the felid trade in the region, emphasizing the interconnectedness of various actors and countries involved. At the same time, we highlight the critical need for more in-depth studies and consistent monitoring to gain a deeper understanding of the dynamics of this trade. Only with such detailed insights will it be possible to design and implement effective behavior change campaigns and targeted interventions to address this issue comprehensively L.599-606.

Lines 533 – 534: I would think, per your findings around corruption, that your findings don’t actually highlight the importance of enforcement. I suggest removing this/rewording to focus only on monitoring and how it could be helpful.
Please, see answer above in the comment of lines 530-533 and new conclusion.

Lines 535 – 537: This is great and true, but a missing piece of this article is your suggestion of how that could be accomplished, based on your data. This is where thematic analysis can play a really crucial role.
Please, see answer above in the comment of lines 530-533 and new conclusion.

Reviewer 2 Report

Comments and Suggestions for Authors

I very much enjoyed this paper and I have no doubt that it is an important contribution and will lead the researchers to either conduct further work (as discussed in the paper) or enable others to do so.

I do have a couple of points for consideration by the researchers as set out below:

Did you gain a favourable ethical opinion for your study. This should be included in the paper, together with the Ethics Committee reference number.

Specific comments

I found this to be a very interesting paper with important insights in to the illegal trade in cats in Cote d'Ivorie. 

Line 46 - amend to say, billions of dollars

Line 82 - do you mean regions when you say populations? Can you say where they are listed as critically endangered please - Red List?

Line 409 - as you have used a percentage elsewhere, you should do so here.

Line 489 - this line does not make sense. Cut "importantly"?

Author Response

General comment

I very much enjoyed this paper and I have no doubt that it is an important contribution and will lead the researchers to either conduct further work (as discussed in the paper) or enable others to do so.

Thank you very much for your positive comments on our manuscript. We are delighted that you found the paper interesting and valuable. Below, we provide detailed responses to your comments and indicate the corresponding revisions in the manuscript.

Specific comments
"Did you gain a favourable ethical opinion for your study? This should be included in the paper, together with the Ethics Committee reference number."
Thank you for highlighting this point. Ethical clearance was obtained from the University of Cape Town as stated in the "Institutional Review Board Statement" section at the end of the manuscript. However, to ensure clarity, we have now included a reference to this approval in the "Ethical considerations and participant consent" subheading in the Methods section.

"Line 46 - amend to say, billions of dollars."
We have revised Line 46 to include "billions of dollars," as suggested. Thank you for pointing this out.

"Line 82 - Do you mean regions when you say populations? Can you say where they are listed as critically endangered, please - Red List?"
We have modified the text to clarify this point by changing "populations" to "sub-populations," as defined by the IUCN (Subpopulations are defined as geographically or otherwise distinct groups in the 
population between which there is little demographic or genetic exchange (typically one 
successful migrant individual or gamete per year or less)). We also added a reference to the IUCN Red List to specify where these sub-populations are listed as critically endangered.

"Line 409 - As you have used a percentage elsewhere, you should do so here."
We agree with this suggestion and have added the corresponding percentage for consistency.

"Line 489 - This line does not make sense. Cut 'importantly'."
Thank you for pointing this out. We have removed "importantly" to improve the clarity of the sentence.

We hope these revisions address your concerns. Thank you once again for your valuable feedback, which has helped improve the quality of our manuscript.

Round 2

Reviewer 1 Report

Comments and Suggestions for Authors

Author Response

I notice the simple summary provides some concrete recommendations, but these same recommendations are not presented in much detail (or at all) in the Discussion and Conclusions. I suggest the authors return to their Discussion and Conclusions and think through how to make concrete recommendations on the basis of their findings. For example, I noted that the authors wrote quite a bit around enforcement reform in their response to me, but I don’t see that inclusion in the manuscript itself, even through reform is called out in the Simple Summary.

Thank you for this insightful comment. After carefully reviewing the Discussion section, we identified the concrete recommendations already present in the text (in green) and acknowledged areas that needed further elaboration. We have made changes to address your comment and ensure these recommendations are explicitly detailed and better integrated into the Discussion and Conclusions sections. All new or revised content is highlighted in blue below.

For further study on the trade, we recommend expanding the scope and designing the study both statistically and geographically to provide more comprehensive data on the scale of the trade in Côte d'Ivoire. Additionally, we suggest conducting long-term tracking and interviews with volunteer sellers, carried out by local and Hausa-speaking researchers. Such interviews could help bypass some of the biases inherent in this study and improve knowledge on trade routes, frequency, and seller profiles. However, due to the illegal nature of this trade, building trust—even with native speakers—will remain a significant challenge for such studies. L509-516

We recommend conducting a large-scale end-user survey, designed as a consumer sur-vey, to gain deeper insights into the profiles of end users and better inform targeted behavioral change campaigns. Such a survey would also help determine whether practitioners are driving the demand or if it is primarily consumer-led. L530-534

"Additionally, our investigation identified the presence of felid products in craft markets and their use in local chiefdoms, practices also documented in the literature [8,13]. However, these environments were not directly investigated in this study, and we recommend conducting targeted research in these contexts." L546-548

"This connection suggests that the Dozos could be key partners in behavior-change campaigns aiming at limiting the use of felid products, as their influence on Ivorian society might help facilitate shifts in traditional practices." L553-557

"Collaborating with cultural leaders would allow to identify and promote acceptable symbolic alternatives to felid skins, ensuring cultural traditions are respected while protecting wildlife. Follow-up research could explore other viable alternatives beyond those identified." L568-571

"To further combat corruption, we recommend implementing the ECOWAS R12 directive, which calls for a regional strategy to tackle wildlife crime by identifying priorities and harmonizing national and regional efforts [52]. Strengthening controls, such as increasing checkpoints along key routes like the Korhogo-Abidjan corridor and borders with Burkina Faso. Most importantly, enhancing training for enforcement agencies is essential, as, in informal discussions, we noticed a lack of awareness among law enforcement personnel about the importance of protecting wild cat species. References suggest that training programs can effectively sensitize individuals to the significance of wildlife conservation and reduce their susceptibility to corruption (Transparency International, 2010). However, these efforts must be paired with governance reforms aimed at reducing corruption.  L.581-591

"We recommend developing a regional database of lion and leopard skin samples for genetic analysis to identify the geographic origins of trafficked items." L595-596

Below I include a few other small points for the authors to address.

Line 228: What were these pre-defined thematic categories, and why did you define them prior to collecting your data? (What criteria did you use to set them – what information did you already have?)

We have added the thematic categories into the methods section (Lines 228–233) for clarity. These categories, including Alternatives or importance, Corruption and controls, Definition of practices, Price information, Origins, Uses, and Other interesting information, were chosen because they reflect recurring themes in the literature, were identified as relevant during initial discussions with authorities and NGOs, and align with findings from similar studies. This approach ensures that the analysis is both comprehensive and grounded in established research.

Line 478: What do you mean by “transhumant”?

We have replaced the term “transhumant” with “nomadic herders” to improve clarity. The word “transhumant” is commonly used in French to describe herders who move seasonally with their livestock, but it may not be as widely understood in English.

 Line 511: Do you mean, “where these are not accessible”? I’m not clear on the meaning of this.

Thank you for pointing this out. We have modified the sentence to clarify the intended meaning. The revised text now expresses that western medicine is often "not accessible" due to higher costs as well as limited availability of drugstores and certain medicines, as noted in the bibliography. L530-533

Line 537 – 538: No need for “although”

We have removed the word "although" from Lines 546–547, as suggested by the reviewer, to improve clarity.

 Line 604: I’m not convinced that you can claim, on the basis of your results, that practitioners are “driving” demand. Do you have evidence that they are encouraging non-users to use felid parts? A stronger argument could be made on the basis of availability of felid products creating the right conditions for demand to appear – but you don’t present evidence that practitioners are driving this demand. I also suggest removing “prevalence of” because it is unnecessary

Thank you for this relevant comment. We have added some testimony in the Results (L.354-362) section to support the hypothesis that practitioners may drive demand. However, we agree with the reviewer that this remains an unproven hypothesis, as we do not have sufficient evidence to substantiate this claim fully. Further investigations, such as end-user surveys, would be necessary to provide more robust data, as outlined in the recommendations added in lines 530-534. Additionally, we have revised the Conclusion section accordingly. Thank you for bringing this to our attention.

Additionally, we have modified the method for generating the heatmap in Figure 3. Instead of transforming lines into points to calculate the heatmap, we directly calculated the density using the line density function in QGIS. The methods section has been updated accordingly (Lines 246–247), and we have provided a new map. While the result is almost the same, it is more precise, even if it has a slightly less appealing "raster-like" appearance. Additionally, we have added the principal road network of West Africa to the map, as we believe it provides important contextual information.